# Prediction of Primary Tumour and Axillary Lymph Node Response to Neoadjuvant Chemo(Targeted) Therapy with Dedicated Breast [18F]FDG PET/MRI in Breast Cancer

**DOI:** 10.3390/cancers15020401

**Published:** 2023-01-07

**Authors:** Cornelis M. de Mooij, Thiemo J. A. van Nijnatten, Briete Goorts, Loes F. S. Kooreman, Isabel W. M. Raymakers, Silke P. L. van Meijl, Maaike de Boer, Kristien B. M. I. Keymeulen, Joachim E. Wildberger, Felix M. Mottaghy, Marc B. I. Lobbes, Marjolein L. Smidt

**Affiliations:** 1Department of Radiology and Nuclear Medicine, Maastricht University Medical Centre, P. Debyelaan 25, 6229 HX Maastricht, The Netherlands; 2Department of Surgery, Maastricht University Medical Centre, P. Debyelaan 25, 6229 HX Maastricht, The Netherlands; 3GROW—School for Oncology and Reproduction, Maastricht University, Universiteitssingel 40, 6229 ER Maastricht, The Netherlands; 4Department of Pathology, Maastricht University Medical Centre, P. Debyelaan 25, 6229 HX Maastricht, The Netherlands; 5Department of Medical Oncology, Maastricht University Medical Centre, P. Debyelaan 25, 6229 HX Maastricht, The Netherlands; 6Department of Nuclear Medicine, RWTH Aachen University Hospital, Pauwelstraße 30, 52074 Aachen, Germany; 7Department of Medical Imaging, Zuyderland Medical Centre, Dr. H. van der Hoffplein 1, 6162 BG Sittard-Geleen, The Netherlands

**Keywords:** breast neoplasms, fluorodeoxyglucose F18, positron emission tomography, magnetic resonance imaging, neoadjuvant therapy

## Abstract

**Simple Summary:**

Neoadjuvant chemo(targeted) therapy (NCT) can downstage disease burden in breast cancer, allowing less invasive surgery. The ability of sequential hybrid [18F]FDG PET/MRI to predict the final pathologic primary tumour response to NCT in breast cancer was investigated. In addition, the value of sequential hybrid [18F]FDG PET/MRI in predicting axillary response was investigated separately in clinically node-positive breast cancer patients. In this study, final pathologic primary tumour and axillary lymph node response prediction with qualitative or quantitative [18F]FDG PET/MRI after NCT is not reliable. However, combining the relative decrease in [18F]FDG PET and MR imaging variables halfway through NCT improved diagnostic performance, especially in predicting the final pathologic axillary lymph node response. These findings suggest that sequential hybrid [18F]FDG PET/MRI could have complementary value in the early prediction of the final pathologic response to NCT in breast cancer.

**Abstract:**

Background: The aim of this study was to investigate whether sequential hybrid [18F]FDG PET/MRI can predict the final pathologic response to neoadjuvant chemo(targeted) therapy (NCT) in breast cancer. Methods: Sequential [18F]FDG PET/MRI was performed before, halfway through and after NCT, followed by surgery. Qualitative response evaluation was assessed after NCT. Quantitatively, the SUV_max_ obtained by [18F]FDG PET and signal enhancement ratio (SER) obtained by MRI were determined sequentially on the primary tumour. For the response of axillary lymph node metastases (ALNMs), SUV_max_ was determined sequentially on the most [18F]FDG-avid ALN. ROC curves were generated to determine the optimal cut-off values for the absolute and percentage change in quantitative variables in predicting response. Diagnostic performance in predicting primary tumour response was assessed with AUC. Similar analyses were performed in clinically node-positive (cN+) patients for ALNM response. Results: Forty-one breast cancer patients with forty-two primary tumours and twenty-six cases of pathologically proven cN+ disease were prospectively included. Pathologic complete response (pCR) of the primary tumour occurred in 16 patients and pCR of the ALNMs in 14 cN+ patients. The AUC of the qualitative evaluation after NCT was 0.71 for primary tumours and 0.54 for ALNM responses. For primary tumour response, combining the percentage decrease in SUV_max_ and SER halfway through NCT achieved an AUC of 0.78. The AUC for ALNM response prediction increased to 0.92 by combining the absolute and the percentage decrease in SUV_max_ halfway through NCT. Conclusions: Qualitative PET/MRI after NCT can predict the final pathologic primary tumour response, but not the ALNM response. Combining quantitative variables halfway through NCT can improve the diagnostic accuracy for final pathologic ALNM response prediction.

## 1. Introduction

Neoadjuvant chemo(targeted) therapy (NCT) has acquired a well-established role in the treatment of invasive breast cancer [1,2,3,4]. NCT can downstage disease burden in the breast and the axillary lymph nodes (ALNs) [5,6,7]. On average, 22% of breast cancer patients achieve a pathologic complete response (pCR) of the primary tumour (i.e., ypT0/is) and 36% of pathologically proven clinically node-positive (cN+) breast cancer patients achieve a pCR of ALN metastases (ALNM) [7,8]. Accurate prediction of disease progression during NCT or pCR after NCT provides the opportunity for response-guided treatment [9].

Concerning the primary tumour, the accuracy of non-invasive imaging in determining the response to NCT has been investigated extensively. Magnetic resonance imaging (MRI) provides high sensitivity in detecting residual disease (RD), while positron emission tomography with computed tomography (PET/CT) using [18F]-fluorodeoxyglucose ([18F]FDG) has high specificity for the detection of pCR [10,11]. Hybrid [18F]FDG PET/MRI demonstrates complementary performance with the important advantage of combining quantitative [18F]FDG PET and MR imaging variables in a single examination [12].

Concerning ALNMs in cN+ breast cancer patients, the advent of less invasive axillary surgical procedures after NCT has increased the importance of accurate non-invasive axillary response assessment [13,14]. Thus far, non-invasive imaging has not been able to reliably determine axillary response after NCT in pathologically proven cN+ breast cancer patients [15]. The diagnostic performance of sequential hybrid [18F]FDG PET/MRI in axillary response prediction in pathologically proven cN+ breast cancer patients has not yet been investigated.

Therefore, the primary aim of this study was to investigate whether sequential dedicated breast hybrid [18F]FDG PET/MRI could accurately predict pathologic primary tumour response to NCT in breast cancer patients. As a secondary aim, pathologic axillary response prediction in pathologically proven cN+ patients was investigated.

## 2. Materials and Methods

This prospective, single-centre study was approved by the local medical research ethics committee. Requirement for informed consent was waived, since sequential [18F]FDG PET/MRI was clinically evaluated for response during NCT.

### 2.1. Patients

Female patients with histopathologically confirmed primary invasive breast cancer with a primary tumour larger than 2 centimetres and/or ALNM confirmed by tissue sampling, who completed NCT and were planned to undergo breast and axillary surgery, were eligible for inclusion. Exclusion criteria were pregnancy, neoadjuvant hormone monotherapy, presence of distant metastasis at diagnosis, and contraindications for MRI. Consecutive eligible patients were offered [18F]FDG PET/MRI for response evaluation to NCT, in case baseline imaging with MRI or [18F]FDG PET/CT was not yet performed.

### 2.2. Neoadjuvant Chemo(targeted) Therapy Regimens

NCT consisted of 4 cycles of 3-weekly doses of doxorubicin and cyclophosphamide, followed by 4 cycles of 3-weekly doses of docetaxel in cases of oestrogen receptor (ER)-positive and/or human epidermal growth factor receptor 2 (HER2)-positive breast cancer, or 12 cycles of weekly doses of paclitaxel in cases of triple negative (TN) breast cancer (Appendix A). In cases of HER2-positive breast cancer, targeted therapy (trastuzumab with/without pertuzumab) was added to the neoadjuvant treatment regimen.

### 2.3. [18F]FDG PET/MRI

Dedicated breast hybrid [18F]FDG PET/MRI was performed at baseline before NCT (PETMRI-1), halfway through NCT after the first 4 cycles (PETMRI-2), and/or after NCT (PETMRI-3) prior to surgery. All scans were acquired using a 3.0 Tesla integrated PET/MRI system (Biograph mMR; Siemens Healthineers, Erlangen, Germany), following a resting period of 45–60 min after [18F]FDG administration. Prior to an intravenous injection of 2 MBq/kg body weight of [18F]FDG, patients fasted for at least four hours and blood glucose was checked to ensure their levels were below 11 mmol/L. Images were acquired from the diaphragm to the top of the humeral head using a dedicated bilateral 16-channel breast radiofrequency coil (Rapid Biomedical, Rimpar, Germany), while patients were placed in prone position with both arms elevated. A detailed description of the protocol has been described previously and is provided in Appendix A [16].

### 2.4. Image Evaluation

All PET images were evaluated by a final-year resident radiology and nuclear medicine physician (T.N.) with four years of clinical experience in PET imaging, using dedicated software (Syngo.via 6.4, Siemens-Healthcare, Erlangen, Germany). For the quantitative analysis of the primary tumour and ALNM on [18F]FDG PET, a volume of interest (VOI) was placed over the most [18F]FDG-avid component of the primary tumour in the breast or the most [18F]FDG-avid ALN, respectively. Maximum and peak standardised uptake values (SUV_max_ and SUV_peak_, respectively) were automatically measured [17]. Additionally, an isoactivity contour was automatically drawn in the VOI using pre-set margin thresholds and the SUV_mean_, metabolic tumour volume (MTV) and total lesion glycolysis (TLG) were calculated. Lastly, the nodal-to-tumour ratio (NT ratio) was calculated by dividing the SUV_max_ of the most [18F]FDG-avid ALN by the SUV_max_ of the primary tumour [18]. In cases of low [18F]FDG avidity, VOI placement was performed with the use of MR images. For the qualitative evaluation of primary tumour response, the complete response of the primary tumour was defined as [18F]FDG uptake that was indistinguishable from the surrounding tissue [17]. For the qualitative evaluation of axillary response, axillary complete response was defined as no ALN with moderately or very intense [18F]FDG uptake [19].

All MR images were evaluated by a dedicated breast radiologist (M.L.) with thirteen years’ experience in breast imaging, using dedicated software (PACS System Sectra Workstation IDS7, version 23.1.10, Sectra Group, Linköping, Sweden). For the quantitative analysis of the primary tumour, the longest diameter (LD) was defined as the maximal diameter of an enhancing lesion measured at peak enhancement in any plane, including intervening areas of non-enhancing tissue. Additionally, a researcher (C.M.) dedicated to breast imaging determined the signal enhancement ratio (SER) and apparent diffusion coefficient (ADC) values. For SER measurements, a circular region of interest (ROI) of 5 mm in diameter was placed on the most enhancing part of the primary tumour at peak enhancement. SER was calculated using the following equation: *SER* = (*S*_1_ – *S*_0_)/(*S*_2_ – *S*_0_), where S_0_, S_1_ and S_2_ represent the signal intensities on pre-contrast, early post-contrast, and late post-contrast images, respectively [20]. For the ADC measurements, a single ROI was manually drawn on the DW images at *b* = 1000 s/mm^2^ on a region with hyperintensity and relatively low ADC to include the entire tumour in the axial slice where the tumour was the largest, avoiding normal breast parenchyma, fat and regions of high T2 signal (e.g., seroma and necrosis) [21]. In primary tumours without residual enhancement on T1W or hyperintensity on DW images halfway through or after NCT, ROIs were placed in the same tissue region as the prior examination. For the qualitative evaluation of primary tumour response, complete response was defined as the absence of residual enhancing tissue. For the qualitative evaluation of axillary response, all the visible ALNs were evaluated using characteristics of suspicious ALNs, including irregular margins, inhomogeneous cortex, perifocal edema, and absence of fatty hilum or chemical shift artifact [22,23]. Axillary complete response was defined as the absence of ALNs with suspicious characteristics.

### 2.5. Pathologic Response Reference Standard

Pre-treatment core needle biopsies of the primary tumour were used for histological subtyping and grading. Tumours were considered positive for ER or progesterone receptor (PR) if at least 10% of cells showed nuclear staining. HER2 positivity was defined as either a score of 3+ following immunohistochemical (IHC) staining or HER2 gene amplification by fluorescent in situ hybridisation. Grading was performed according to the modified Bloom–Richardson system.

Post-treatment surgical specimens of the breast and the axilla were used to evaluate the response. Histopathological measurement of residual tumour size was performed during grossing and was later correlated microscopically. Primary tumour pCR was defined as the absence of residual invasive cancer in the breast after NCT (ypT0/is). Axillary pCR was defined as the absence of tumour cells or isolated tumour cells (≤0.2 mm or less than 200 cells). Residual axillary disease was defined as the presence of micrometastases (>0.2 and ≤2.0 mm) and/or macrometastases (>2.0 mm).

Histopathological analyses were performed in accordance with the Dutch national breast cancer guideline at the time of diagnosis [24].

### 2.6. Statistical Analysis

The absolute values of all quantitative [18F]FDG PET/MR imaging variables at each time point, as well as the percentage decrease halfway through and after NCT, were compared between patients with a primary tumour and axillary pCR and RD separately by means of the Mann–Whitney U test. For all the significant quantitative variables, receiver operating characteristic (ROC) curves were generated to determine the cut-off value with optimal sensitivity and specificity. Diagnostic performance, expressed as sensitivity, specificity, positive predictive value (PPV), negative predictive value (NPV) and area under the curve (AUC) with 95% confidence intervals (95% CIs), was calculated for each significant quantitative imaging variable at the optimal cut-off, both separately and combined. Lastly, the diagnostic performance of qualitative [18F]FDG PET, MRI, and [18F]FDG PET/MRI after NCT were calculated. For all analyses, the detection of residual disease via imaging or pathology analysis was considered as positive and pCR via imaging or pathology analysis was considered as negative. A two-sided *p*-value of <0.05 was considered to be statistically significant. R project software (version 4.2.0, R Foundation for Statistical Computing, Vienna, Austria) was used to perform the statistical analyses.

## 3. Results

### 3.1. Clinicopathologic Characteristics

Between February 2015 and July 2017, 41 breast cancer patients with 42 primary tumours and 26 cN+ axillae were included in this prospective study (Table 1). Primary tumour response evaluation halfway through and after completion of NCT was performed in 38 and 37 patients, and axillary response evaluation in 22 and 21 patients, respectively (Figure 1).

### 3.2. Quantitative Imaging Variables in Relation to Response

In primary tumour pCR, the percentage decrease in SUV_max_ (−82.6 vs. −40.7, *p* = 0.017) and SER (−30.1 vs. −13.0, *p* = 0.044) halfway through NCT was significantly higher than in primary tumour RD (Table 2, Figure 2). After NCT, in primary tumour pCR, the median LD was significantly lower (0.0 vs. 15.0, *p* = 0.018) and the percentage decrease in LD (−100.0 vs. −40.9, *p* = 0.012) and SER (−54.3 vs. −38.4, *p* = 0.013) were significantly higher than in primary tumour RD. No differences were reported for MTV and TLG at any of the thresholds, neither in the absolute values nor in the percentages decrease at any time point (Appendix A).

In axillary pCR, the median SUV_max_ of the most [18F]FDG-avid ALN (0.5 vs. 0.9, *p* = 0.030) and NT ratio (0.4 vs. 0.6, *p* = 0.041) halfway through NCT was significantly lower than in axillary RD. The percentage decrease in SUV_max_ (−88.0 vs. −59.8, *p* = 0.010) and NT ratio (−59.7 vs. −35.7, *p* = 0.018) halfway through NCT was significantly higher in axillary pCR than in axillary RD (Table 3, Figure 3). After NCT, in axillary pCR, the median primary tumour LD was significantly lower (0.0 vs. 15.0, *p* = 0.047) and its percentage decrease (−100.0 vs. −53.8, *p* = 0.026) was significantly higher than in axillary RD (Appendix A).

### 3.3. Response Prediction

The diagnostic performance for the prediction of pathologic primary tumour response is summarised in Table 3. After NCT, the result of the qualitative response evaluation was very similar to [18F]FDG PET and MRI with an AUC of 0.67 (95% CI 0.49–0.85), which only marginally improved by means of [18F]FDG PET/MRI consensus to 0.71 (95% CI 0.53–0.89). Combining the percentage decrease in SER and SUV_max_ of the primary tumour halfway through NCT achieved an AUC of 0.78 (95% CI 0.63–0.93).

Table 4 depicts the diagnostic performance for prediction of axillary response. After NCT, the result of the qualitative response evaluation for [18F]FDG PET, MRI and [18F]FDG PET/MRI consensus was poor, with AUCs of 0.50 (95% CI 0.23–0.77), 0.54 (95% CI 0.27–0.80), and 0.54 (95% CI 0.27–0.80), respectively. Combining the absolute SUV_max_ of the most [18F]FDG-avid ALN halfway through NCT with its percentage decrease achieved an AUC of 0.92 (95% CI 0.79–1.00).

## 4. Discussion

The aim of this study was to investigate the diagnostic accuracy of sequential [18F]FDG PET/MRI in predicting primary tumour and ALNM response to NCT. For final pathologic primary tumour response prediction, combining the decrease in SUV_max_ and SER of the primary tumour halfway through NCT can improve the value of [18F]FDG PET/MRI, compared to the qualitative evaluation after NCT. In addition, we found that combining [18F]FDG PET and MRI does not improve the diagnostic accuracy of qualitative primary tumour response evaluation after NCT. For final pathologic ALNM response prediction in cN+ breast cancer patients, combining the absolute SUV_max_ measured on the most [18F]FDG-avid ALN halfway through NCT with its relative decrease can accurately predict axillary response. Based on the findings of this study, predicting axillary response with [18F]FDG PET/MRI after NCT is inadequate and does not justify its use.

The diagnostic performance of qualitative primary tumour response evaluation with [18F]FDG PET/MRI was similar to the separate evaluation of [18F]FDG PET and MRI, as indicated by similar AUCs, but displayed improved sensitivity and NPV when combining modalities. This complementary effect can be explained by RD that is either morphologically normalised with residual metabolic activity or has morphological abnormalities without residual metabolic activity. The diagnostic performance of separate qualitative [18F]FDG PET and MRI in detecting primary tumour response after NCT in this study is slightly lower compared to the pooled estimates of [18F]FDG PET/CT and MRI reported in several meta-analyses [25,26]. Similar to our results, the specificity of [18F]FDG PET/CT and MRI after NCT is often low and previous studies have reported that this inability to detect pCR could be explained by NCT-induced inflammation, sclerosis, necrosis, perilesional edema and the presence of ductal carcinoma in situ (DCIS) [27].

Sekine et al. previously investigated the diagnostic performance of [18F]FDG PET/MRI in detecting primary tumour response after NCT in breast cancer patients [28]. In a cohort of 74 patients, their similar qualitative approach achieved a sensitivity and specificity for primary tumour response of 72.2% and 78.6%, respectively. The diagnostic performance heavily depended on MRI, since the majority of the patients demonstrated metabolic activity normalised to background on [18F]FDG PET, regardless of response. This is contrary to our results, since we report similar performances for qualitative [18F]FDG PET and MRI. Lastly, Sekine et al. only investigated a qualitative approach and did not include quantitative [18F]FDG PET or MR imaging variables. Interestingly, Sekine et al. found striking differences in diagnostic performance between different breast cancer subtypes.

Increasing evidence indicates that breast cancer subtypes present differently on 18F-FDG PET and MRI, indicated by the significant differences between subtypes in qualitative and quantitative imaging variables [29,30]. These significant differences between subtypes are not limited to baseline, but extend to different patterns of response in non-invasive imaging, which impacts the accuracy of detecting or predicting pathological primary tumour or axillary response [31,32,33]. Consequently, the prediction of pathological primary tumour or axillary response to NCT with non-invasive imaging could benefit from subtype-specific cut-off values for quantitative imaging variables. In addition, differences in the response patterns on MRI also seem to differ between different breast cancer subtypes [34]. Unfortunately, the small sample size in this preliminary study did not permit an analysis per subtype.

Our results suggest that the diagnostic performance in predicting primary tumour response can be improved with quantitative [18F]FDG PET/MR imaging variables. Similar to previous results, the percentage decrease in SUV_max_ halfway through NCT strongly improves sensitivity [10,26]. Interestingly, using a cut-off for the quantitative MR imaging variable LD improved specificity and PPV compared to the qualitative evaluation, possibly by correctly identifying residual enhancement caused by inflammation that is reactive to NCT or DCIS as pCR [10,27]. ADC was not predictive of response to NCT in our cohort of patients and a recent systematic review reports high heterogeneity regarding the clinical and technical aspects of DWI for response prediction [35]. The complementary value of [18F]FDG PET/MRI is mainly established by combining the percentage decrease in SER and SUV_max_ halfway through NCT, which strongly improved specificity and PPV. This is in line with studies that combine quantitative imaging variables from separate [18F]FDG PET/CT and MRI, as well as from [18F]FDG PET/MRI [12,36,37].

Similar to our results, two studies reported improved early primary tumour response prediction using quantitative [18F]FDG PET/MRI. Cho et al. achieved maximum sensitivity by combining the [18F]FDG PET/MR imaging variables SER and TLG_30%_ [36]. In contrast, none of the volumetric [18F]FDG PET parameters in this study were found to be associated with response, possible due to inaccurate delineation in cases of response, since a decrease in SUV_max_ paradoxically increases the volume when using automatic isoactivity contouring. Cho et al. performed a second examination after one cycle of NCT and defined pCR as the absence of invasive cancer and DCIS in both the breast and ALNs. Additionally, Cho et al. did not find the percentage decrease in SUV_max_ to be predictive of response, possibly due to their small sample size. In a cohort of 14 patients, Wang et al. found the combination of percentage decrease in ADC_min_ after one or two cycles of NCT with either SUV_max_ or TLG_40%_ to be best predictive of response [37]. However, Wang et al. included proton magnetic resonance spectroscopy after DCE-MRI for VOI placement and defined pCR as less than 10% residual cellularity of invasive cells, indicating potential residual cancer in cases with pCR.

The qualitative assessment of axillary response after NCT with dedicated breast [18F]FDG PET/MRI in this study is poor, due to the normalisation of the majority of ALNs on imaging. Sensitivity and PPV of the separate evaluation of [18F]FDG PET are considerably worse compared to the pooled estimates of three primary studies in a recent meta-analysis [15]. However, two of these studies evaluated morphologic criteria for CT and only a decrease in tumour deposit in the ALN was found to be predictive of response in a study by You et al. [38,39]. Similar to our methods, Garcia Vicente et al. evaluated [18F]FDG PET and achieved a sensitivity and PPV of 37% and 68%, respectively [40]. In their study, SUV_max_ measured on the most [18F]FDG-avid ALN ranged up to 13.2, which is considerably higher than the maximum SUV_max_ of 1.3 in our study, possibly explaining the low sensitivity. Regarding the separate evaluation of MRI in this study, the poor sensitivity and PPV cannot be entirely explained. However, Hieken et al. reports metastasis with diameters as large as 12 mm among their false-negative cases, with extranodal extension present in half of these patients [38].

While none of the patients with axillary RD were correctly identified by the qualitative assessment after NCT, the prediction of response halfway through NCT with quantitative [18F]FDG PET imaging variables resulted in sensitivities ranging from 83 to 100%. Combining the absolute decrease with the percentage decrease in SUV_max_ measured on the most [18F]FDG-avid ALN halfway through NCT, a maximum sensitivity or specificity, and thus PPV and NPV, can be achieved. This is in line with the results of two previous studies in which axillary response prediction based on the percentage decrease in SUV_max_ was more accurate early during NCT after one to three courses, compared to after NCT [41,42]. The association of primary tumour LD with axillary response can most likely be attributed to the established correlation between primary tumour and axillary response [43]. Similarly, Eun et al. also found the decrease in primary tumour size to be predictive of axillary response during, as well as after, NCT [44].

Our study has some limitations. The number of included patients was relatively small, especially the cN+ subgroup of patients. As a consequence, we report wide 95% CIs for the diagnostic performance, which should be interpreted with this limitation in mind. Second, the small sample size hindered the separate evaluation of breast cancer subtypes with regard to qualitative and quantitative response evaluation, while it is known to influence diagnostic performance. Third, the interobserver variability of quantitative [18F]FDG PET and MR imaging variables is not assessed in this study. However, Cho et al. has previously reported reliable reproducibility for similar variables [36]. Lastly, we included DCIS as a primary tumour pCR, which could influence the diagnostic performance.

## 5. Conclusions

The complementary value of hybrid dedicated breast [18F]FDG PET/MRI in primary tumour response detection is mainly established by combining quantitative imaging variables. For axillary response prediction in cN+ breast cancer patients, combining the absolute SUV_max_ of the most [18F]FDG-avid ALN halfway through NCT with its percentage decrease strongly improved the diagnostic performance of [18F]FDG PET/MRI. Based on the findings of this study, the diagnostic performance in predicting axillary response with [18F]FDG PET/MRI after NCT is insufficient and does not justify its use at this time-point.

## Figures and Tables

**Figure 1 cancers-15-00401-f001:**
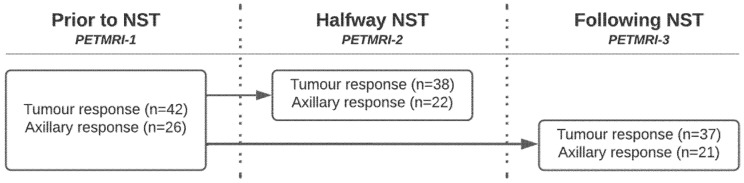
Number of patients per [18F]FDG PET/MRI timepoint included in primary tumour and axillary response evaluation.

**Figure 2 cancers-15-00401-f002:**
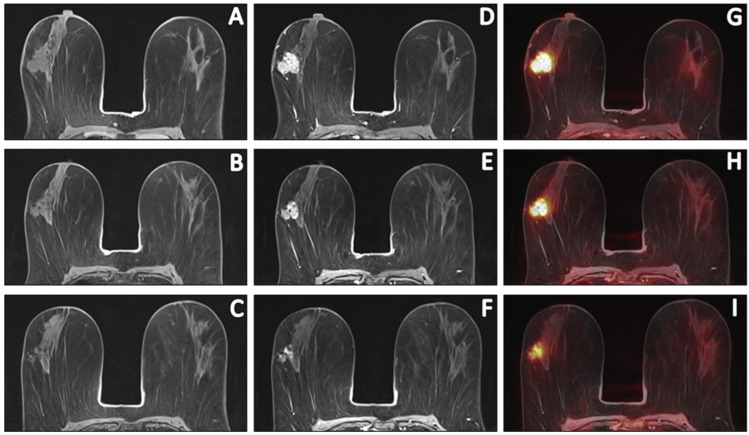
Sequential [18F]FDG PET/MR images of a 58-year-old patient with invasive carcinoma of no specific type (ER+/HER2−, grade 2) who had persistent RD (ypT1c; 16 mm) after completion of NCT. Pre-contrast T1W (**A**–**C**), post-contrast T1W at peak enhancement (**D**–**F**) and fusion of post-contrast images with PET (**G**–**I**) images depict the primary tumour response prior to, halfway through and after NCT. Comparing pre- and post-contrast T1W images revealed a primary tumour with an LD of 29 mm and evaluation of PET images depicted an [18F]FDG-avid tumor with an SUV_max_ of 13.59. Halfway through NCT, the LD and SUV_max_ decreased to 28 mm and 8.05, respectively. After NCT, the LD and SUV_max_ decreased to 25 mm and 4.70, respectively. This patient was considered a true-positive case.

**Figure 3 cancers-15-00401-f003:**
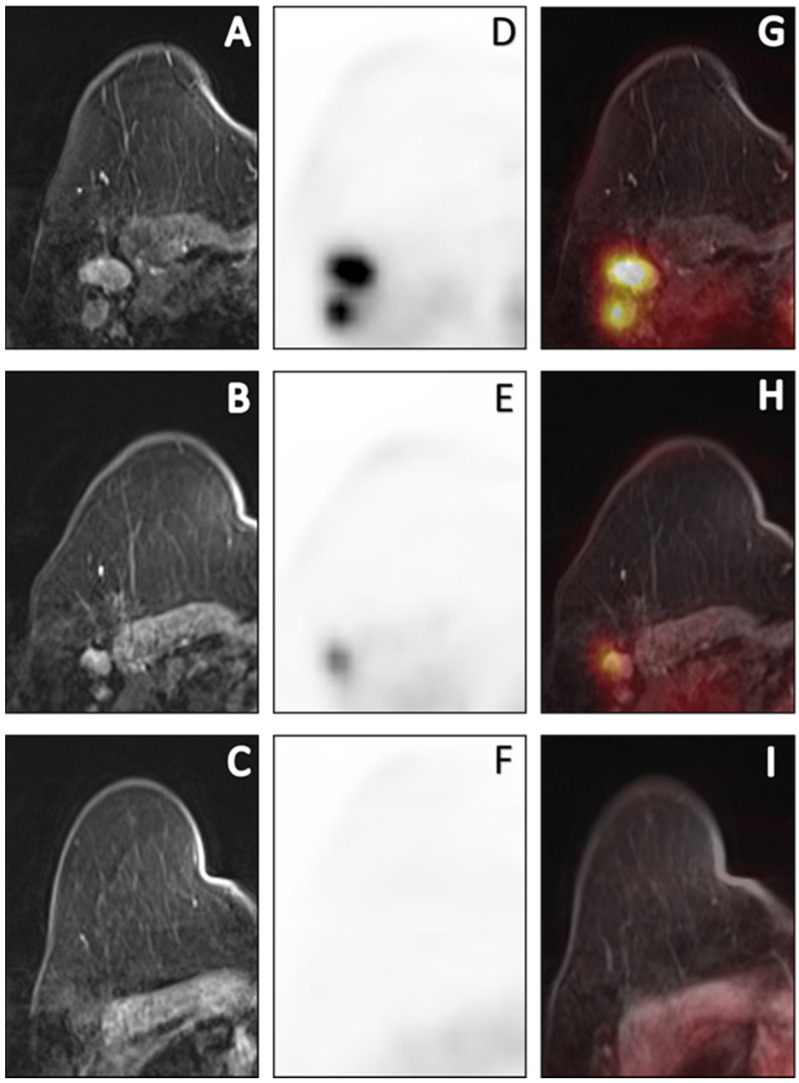
Sequential [18F]FDG PET/MR images of a 69-year-old cN+ patient with invasive carcinoma of no specific type (ER+/HER2-, grade 2) who had persistent axillary disease (ypN1, 3 macrometastases) after completion of NCT. Post-contrast T1W at peak enhancement (**A**–**C**), PET (**D**–**F**) and fusion of post-contrast images with PET (**G**–**I**) images depict the axillary response prior to, halfway through and after NCT. Prior to NCT, a total of 3 PET-positive nodes reaching a maximum SUV_max_ of 7.35 were depicted. Halfway through NCT, SUV_max_ decreased by 61% to 1.78. After completion of NCT, metabolic dissolution in the entire axillary region without any suspicious nodes on MRI was found. Based on the qualitative evaluation after NCT, this was a false-negative case. Using quantitative imaging variables halfway through NCT, this patient was correctly predicted to have residual axillary disease.

**Table 1 cancers-15-00401-t001:** Clinicopathologic and operative characteristics in relation to final pathologic response of all included patients.

	Primary Tumour Response in All Patients			Axillary Response in cN+ Patients		
Characteristics	Total	pCR	RD	Total	pCR	RD
	(*n* = 42)	(*n* = 16)	(*n* = 26)	(*n* = 26)	(*n* = 14)	(*n* = 12)
**Age (years)**						
Median, range	50 (32–69)	48 (36–59)	51 (32–69)	49 (32–69)	48 (32–69)	52 (37–69)
**Clinical tumour size (mm)**						
Median, range	34 (13–78)	46 (13–78)	31 (13–77)	38 (13–78)	38 (13–78)	36 (16–70)
**Clinical T status**						
cT1	5 (11.9)	2 (12.5)	3 (11.5)	4 (15.4)	2 (14.3)	2 (16.7)
cT2	27 (64.3)	8 (50.0)	19 (73.1)	15 (57.7)	7 (50.0)	8 (66.7)
cT3	9 (21.4)	6 (37.5)	3 (11.5)	7 (26.9)	5 (35.7)	2 (16.7)
cT4	1 (2.4)	0 (0.0)	1 (3.8)	0 (0.0)	0 (0.0)	0 (0.0)
**Clinical N status**						
cN0	15 (35.7)	6 (37.5)	9 (34.6)	0 (0.0)	0 (0.0)	0 (0.0)
cN1	24 (57.1)	8 (50.0)	16 (61.5)	24 (92.3)	13 (92.9)	11 (91.7)
cN2	1 (2.4)	0 (0.0)	1 (3.8)	0 (0.0)	0 (0.0)	0 (0.0)
cN3	2 (4.8)	2 (12.5)	0 (0.0)	2 (7.7)	1 (7.1)	1 (8.3)
**Focality**						
Unifocal	26 (61.9)	9 (56.3)	17 (65.4)	13 (50.0)	6 (42.9)	7 (58.3)
Multifocal	2 (4.8)	1 (6.3)	1 (3.8)	1 (3.8)	1 (7.1)	0 (0.0)
Multicentric	4 (9.5)	4 (25.0)	0 (0.0)	3 (11.5)	2 (14.3)	1 (8.3)
Multicentric/multifocal	10 (23.8)	2 (12.5)	8 (30.8)	9 (34.6)	5 (35.7)	4 (33.3)
**ER status**						
Negative	16 (38.1)	8 (50.0)	8 (30.8)	10 (38.5)	6 (42.9)	4 (33.3)
Positive	26 (61.9)	8 (50.0)	18 (69.2)	16 (61.5)	8 (57.1)	8 (66.7)
**PR status**						
Negative	26 (61.9)	12 (75.0)	14 (53.8)	17 (65.4)	9 (64.3)	8 (66.7)
Positive	16 (38.1)	4 (25.0)	12 (46.2)	9 (34.6)	5 (35.7)	4 (33.3)
**HER2 status**						
Negative	29 (69.0)	6 (37.5)	23 (88.5)	20 (76.9)	8 (57.1)	12 (100.0)
Positive	13 (31.0)	10 (62.5)	3 (11.5)	6 (23.1)	6 (42.9)	0 (0.0)
**Subtype**						
ER+/HER2−	19 (45.2)	2 (12.5)	17 (65.4)	13 (50.0)	5 (35.7)	8 (66.7)
ER+/HER2+	7 (16.7)	6 (37.5)	1 (3.8)	3 (11.5)	3 (21.4)	0 (0.0)
ER−/HER2+	6 (14.3)	4 (25.0)	2 (7.7)	3 (11.5)	3 (21.4)	0 (0.0)
TNBC	10 (23.8)	4 (25.0)	6 (23.1)	7 (26.9)	3 (21.4)	4 (33.3)
**Tumour grade (mBR)**						
Grade 1	4 (9.5)	1 (6.3)	3 (11.5)	4 (15.4)	1 (7.1)	3 (25.0)
Grade 2	22 (52.4)	8 (50.0)	14 (53.8)	13 (50.0)	6 (42.9)	7 (58.3)
Grade 3	16 (38.1)	7 (43.8)	9 (34.6)	9 (34.6)	7 (50.0)	2 (16.7)
**Type of breast surgery**						
BCS	25 (59.5)	10 (62.5)	15 (57.7)	15 (57.7)	10 (71.4)	5 (41.7)
Ablatio	17 (40.5)	6 (37.5)	11 (42.3)	11 (42.3)	4 (28.6)	7 (58.3)
**Type of axillary surgery**						
SLNB	17 (40.5)	7 (43.8)	10 (38.5)	1 (3.8)	1 (7.1)	0 (0.0)
RISAS	3 (7.1)	0 (0.0)	3 (11.5)	22 (84.6)	11 (78.6)	11 (91.7)
ALND	22 (52.4)	9 (56.3)	13 (50.0)	3 (11.5)	2 (14.3)	1 (8.3)

Abbreviations: ALND, axillary lymph node dissection; BCS, breast-conserving surgery; cN+, clinically node-positive; ER, oestrogen receptor; HER2, human epidermal growth factor receptor 2; mBR, modified Bloom–Richardson; pCR, pathologic complete response; PR, progesterone receptor; RD, residual disease; RISAS, radioactive iodine seed localisation in the axilla with the sentinel node procedure; SLNB, sentinel lymph node biopsy; TNBC, triple-negative breast cancer.

**Table 2 cancers-15-00401-t002:** Significant imaging variables in relation to final pathologic response.

	pCR		RD		*p*-Value	Optimal Cut-Off
** *Primary tumour response* **						
**SUV_max_ ***						
Δ2-1 (%)	−82.6	(−94.1 to −9.2)	−40.7	(−87.7 to 8.9)	**0.017**	−75.0
**LD**						
PETMRI-3 (mm)	0	(0.0 to 37.0)	15	(0.0 to 38.0)	**0.018**	11
Δ3-1 (%)	−100.0	(−100.0 to −19.57)	−40.9	(−100.0 to 0.0)	**0.012**	−68.4
**SER**						
Δ2-1 (%)	−30.1	(−68.0 to 8.0)	−13.0	(−69.5 to 46.5)	**0.044**	−23.9
Δ3-1 (%)	−54.3	(−75.4 to −14.5)	−38.4	(−65.3 to 0.6)	**0.013**	−52.6
** *Axillary response* **						
**SUV_max_ †**						
PETMRI-2	0.5	(0.4 to 2.7)	0.9	(0.6 to 5.5)	**0.03**	0.58
Δ2-1 (%)	−88.0	(−96.1 to −37.3)	−59.8	(−93.6 to −7.3)	**0.01**	−75.5
**NT ratio**						
PETMRI-2	0.4	(0.2 to 1.5)	0.6	(0.2 to 3.8)	**0.041**	0.42
Δ2-1 (%)	−59.7	(−92.6 to 42.3)	−35.7	(−66.0 to 78.1)	**0.018**	−46.8
**LD**						
PETMRI-3 (mm)	0	(0.0 to 38.0)	15	(0.0 to 30.0)	**0.047**	11
Δ3-1 (%)	−100.0	(−100.0 to −28.3)	−53.8	(−100.0 to 0.0)	**0.026**	−71.1

Quantitative imaging variables are shown as the median and range. Comparison between response groups using Mann–Whitney U test. Symbols: *, similar cut-off values and diagnostic performance for SUV_peak_, SUV_30%_, SUV_40%_, and SUV_50%_; †, similar cut-off value and diagnostic performance for SUV_peak_. Abbreviations: LD, longest diameter; NT ratio, nodal-to-tumour ratio; pCR, pathologic complete response; RD, residual disease; SER, signal enhancement ratio; SUV_max_, maximum standardised uptake value.

**Table 3 cancers-15-00401-t003:** Diagnostic performance in the prediction of pathologic primary tumour.

	Sensitivity	Specificity	PPV	NPV	AUC
** *Qualitative response evaluation* **					
PET-3	**71** (15/21) [48–89]	**62** (10/16) [35–85]	**71** (15/21) [48–89]	**62** (10/16) [35–85]	**0.67** [0.49–0.85]
MRI-3	**71** (15/21) [48–89]	**62** (10/16) [35–85]	**71** (15/21) [48–89]	**62** (10/16) [35–85]	**0.67** [0.49–0.85]
PETMRI-3	**86** (18/21) [64–97]	**56** (9/16) [30–80]	**72** (18/25) [51–88]	**75** (9/12) [43–95]	**0.71** [0.53–0.89]
** *Quantitative response evaluation* **					
**SUV_max_**					
Δ2-1 (%)	**92** (23/25) [74–99]	**62** (8/13) [32–86]	**82** (23/28) [63–94]	**80** (8/10) [44–97]	**0.74** [0.53–0.94]
**LD**					
PETMRI-3 (mm)	**62** (13/21) [38–82]	**81** (13/16) [54–96]	**81** (13/16) [54–96]	**62** (13/21) [38–82]	**0.72** [0.55–0.89]
Δ3-1 (%)	**67** (14/21) [43–85]	**88** (14/16) [62–98]	**88** (14/16) [62–98]	**67** (14/21) [43–85]	**0.74** [0.57–0.90]
**SER**					
Δ2-1 (%)	**64** (16/25) [43–82]	**69** (9/13) [39–91]	**80** (16/20) [56–94]	**50** (9/18) [26–74]	**0.70** [0.52–0.88]
Δ3-1 (%)	**76** (16/21) [53–92]	**62** (10/16) [35–85]	**73** (16/22) [50–89]	**67** (10/15) [38–88]	**0.74** [0.58–0.90]
**Combined variables**					
SUV_max_ or SER (Δ2-1)	**96** (24/25) [80–100]	**38** (5/13) [14–68]	**75** (24/32) [57–89]	**83** (5/6) [36–100]	**0.67** [0.48–0.87]
SUV_max_ and SER (Δ2-1)	**64** (16/25) [43–82]	**92** (12/13) [64–100]	**94** (16/17) [71–100]	**57** (12/21) [34–78]	**0.78** [0.63–0.93]
LD or SER (Δ3-1)	**81** (17/21) [58–95]	**63** (10/16) [35–85]	**74** (17/23) [52–90]	**71** (10/14) [42–92]	**0.72** [0.54–0.89]
LD and SER (Δ3-1)	**62** (13/21) [38–82]	**88** (14/16) [62–98]	**87** (13/15) [60–98]	**64** (14/22) [41–83]	**0.75** [0.59–0.91]

Abbreviations: AUC, area under the receiver operating characteristic curve; LD, longest diameter; NPV, negative predictive value; NT ratio, nodal-to-tumour ratio; PPV, positive predictive value; SER, signal enhancement ratio; SUV_max_, maximum standardised uptake value.

**Table 4 cancers-15-00401-t004:** Diagnostic performance in the prediction of pathologic axillary response.

	Sensitivity	Specificity	PPV	NPV	AUC
** *Qualitative response evaluation* **					
PET-3	**0** (0/7) [0–14]	**100** (14/14) [77–100]	**0** (0/0) [0–0]	**67** (14/21) [43–85]	**0.50** [0.23–0.77]
MRI-3	**0** (0/7) [0–41]	**93** (13/14) [66–100]	**0** (0/1) [0–97]	**65** (13/20) [41–85]	**0.54** [0.27–0.80]
PETMRI-3	**0** (0/7) [0–41]	**93** (13/14) [66–100]	**0** (0/1) [0–97]	**65** (13/20) [41–85]	**0.54** [0.27–0.80]
** *Quantitative response evaluation* **					
**SUV_max_**					
PETMRI-2	**100** (12/12) [74–100]	**60** (6/10) [26–88]	**75** (12/16) [48–93]	**100** (6/6) [54–100]	**0.78** [0.57–0.98]
Δ2-1 (%)	**83** (10/12) [52–98]	**80** (8/10) [44–97]	**83** (10/12) [52–98]	**80** (8/10) [44–97]	**0.83** [0.64–1.00]
**NT ratio**					
PETMRI-2	**92** (11/12) [62–100]	**60** (6/10) [26–88]	**73** (11/15) [45–92]	**86** (6/7) [42–100]	**0.80** [0.60–1.00]
Δ2-1 (%)	**83** (10/12) [52–98]	**80** (8/10) [44–97]	**83** (10/12) [52–98]	**80** (8/10) [44–97]	**0.76** [0.55–0.97]
**LD (mm)**					
PETMRI-3	**71** (5/7) [29–96]	**86** (12/14) [57–98]	**71** (5/7) [29–96]	**86** (12/14) [57–98]	**0.75** [0.50–0.99]
Δ3-1 (%)	**71** (5/7) [29–96]	**86** (12/14) [57–98]	**71** (5/7) [29–96]	**86** (12/14) [57–98]	**0.78** [0.54–1.00]
**Combined variables**					
SUV_max_ (2) or SUV_max_ (Δ2-1)	**100** (12/12) [74–100]	**40** (4/10) [12–74]	**67** (12/18) [41–87]	**100** (4/4) [40–100]	**0.70** [0.47–0.93]
SUV_max_ (2) and SUV_max_ (Δ2-1)	**83** (10/12) [52–98]	**100** (10/10) [69–100]	**100** (10/10) [69–100]	**83** (10/12) [52–98]	**0.92** [0.79–1.00]

Abbreviations: AUC, area under the receiver operating characteristic curve; LD, longest diameter; NPV, negative predictive value; NT ratio, nodal-to-tumour ratio; PPV, positive predictive value; SER, signal enhancement ratio; SUV_max_, maximum standardised uptake value.

## Data Availability

Raw data were generated at the Maastricht University Medical Centre. The authors confirm that the data and the code for data cleaning and analysis of this study are available from the corresponding author upon reasonable request.

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
