# Peer review of "Prediction of Primary Tumour and Axillary Lymph Node Response to Neoadjuvant Chemo(Targeted) Therapy with Dedicated Breast [18F]FDG PET/MRI in Breast Cancer"

_cancers, 2023, doi:10.3390/cancers15020401_

Round 1

Reviewer 1 Report

In the present manuscript, De Mooij et al analyze the role of FDG PET/MRI into predict response to the therapy in BC patients. I would like to congratulate with the authors for the excellent work. 

The paper is well written and presented

Only some minor issues to address

Please improve treatment section. Maybe a table could be useful with treatment and response information

How many patients did perform trastuzumab plus pertuzumab? Do you see any difference into predict treatment response with PET/MRI between chemotherapy and monoclonal antibody based therapy? If yes could you discuss it in the appropriate section?

Reviewer 2 Report

I read with interest the article which analyzes the predictive values of radiological procedures([18F]FDG PET/MRI) dedicated to the metabolic and functional analysis of pathological tissues.

I find the question concerning the outcome of patients undergoing neoadjuvant therapy and the verification of the diagnostic potential of these methods very interesting, not only in terms of surgical orientation but also with a prognostic value. Furthermore, a greater diagnostic accuracy in the middle of the treatment pathway can identify a class of more selected patients and with a probable more favorable prognosis as they are characterized by an earlier metabolic response.

However, I wonder if it could be of further help to dedicate these diagnostic modalities above all to phenotypes with higher glucose metabolism (TN, HER2+, high ki-67, G3). I would try to include in the discussion a starting point of this kind in which the method can be more sensitive and specific.

In this context, please cite the reference below:

Orsaria P, Chiaravalloti A, Caredda E, Marchese PV, Titka B, Anemona L, Portarena I, Schillaci O, Petrella G, Palombi L, Buonomo OC. Evaluation of the Usefulness of FDG-PET/CT for Nodal Staging of Breast Cancer. Anticancer Res. 2018 Dec;38(12):6639-6652. doi: 10.21873/anticanres.13031. Erratum in: Anticancer Res. 2019 Jan;39(1):527. PMID: 30504372.

I recommend this article with this minor review..

I found it well written, the English is good and the text is clear.

Even if the numbers are not high, the techniques are innovative,  and it is a really interesting study.

I recommend this article with this minor review.

Posttreatment surgical specimens of the breast and the axilla were used to evaluate 163

The manuscript clear, relevant for the field and presented in a well-structured manner.

The cited references mostly recent publications (within the last 5 years) are relevant.

The manuscript scientifically sound and is the experimental design appropriate to test the hypothesis.

The manuscript’s results reproducible based on the details given in the methods section.

The figures/tables/images/schemes are appropriate and they properly show the data( easy to interpret and understand)

The statistical analysis or data acquired are appropriate.

The conclusions are consistent with the evidence and arguments presented the ethics statements and data availability statements to ensure they are adequate.

In conclusion the question original and well-defined and the results provide an advancement of the current knowledge. In this regard the work fit the journal scope. The results interpreted appropriately and are significant. All conclusions are justified and supported by the results and the hypotheses carefully identified. The article written in an appropriate way and the data and analyses presented appropriately.

The conclusions interesting for the readership of the journal and will the paper attract a wide readership.

Reviewer 3 Report

- Overall, I'd avoid the use of terms such as "similar analysis" or "generally", especially when dealing with the methods used. 

- What do the Authors mean by "halfway NCT"? Since treatment varied according to BC molecular subtype, which were the exact time points for imaging assessment according to the different NCT schedules?

- On which plane was the maximum diameter measurement performed?

- Please add a reference for pathological reference standard

- I'd add in the "image analysis paragraph" that percentages decrease of quantitative parameters were also calculated 

- Considering the prospective nature of this study, info on the patient population can be directly reported in MM section

- Table 1: p values of the comparison of clinical data between the two groups should be reported

- Could you please add some info on informed consent waived by the institutional research committee in this prospective investigation?
